# Two Members of Vitamin-K-Dependent Proteins, Gla-Rich Protein (GRP) and Matrix Gla Protein (MGP), as Possible New Players in the Molecular Mechanism of Osteoarthritis

**DOI:** 10.3390/jcm13175159

**Published:** 2024-08-30

**Authors:** Burhan Kurtulus, Numan Atilgan, Mehmet Yilmaz, Recep Dokuyucu

**Affiliations:** 1Department of Orthopedics and Traumatology, Ankara Diskapi Yildirim Beyazit Education and Research Hospital, Ankara 06110, Turkey; kurtulusburhan@gmail.com; 2Department of Hand Surgery, Private Clinic, Gaziantep 27000, Turkey; doktor_dao@hotmail.com; 3Department of Orthopedic Surgery, Gaziantep City Hospital, Gaziantep 27060, Turkey; doctor_yilmaz@hotmail.com; 4Department of Physiology, Medical Specialization Training Center (TUSMER), Ankara 06420, Turkey; 5Physioclinic Private Clinic, Gaziantep 27090, Turkey

**Keywords:** matrix Gla protein, Gla-rich protein, synovial fluid, osteoarthritis

## Abstract

**Objectives:** The pathophysiology of osteoarthritis is mainly unknown. Matrix Gla protein (MGP) and Gla-rich protein (GRP) are both vitamin-K-dependent mineralization inhibitors. In this study, we aimed to compare the levels of MGP and GRP in the synovial fluid of osteoarthritic (OA) and non-osteoarthritic (non-OA) knee joints. **Materials and Methods:** Two groups were formed, with one consisting of patients with OA and the other non-OA, serving as a control group. The non-OA group included individuals who had arthroscopic surgery for non-cartilage-related issues. In the OA group, all participants had undergone total knee arthroplasty because of grade 4 primary degenerative osteoarthritis. During the operation, at least 1 mL of knee synovial fluid was collected. The GRP and MGP levels in the synovial fluid were measured using an ELISA kit. **Results:** The mean age in the OA group (62.03 ± 11.53 years) was significantly higher than that in the non-OA group (47.70 ± 14.49 years; *p* = 0.0001). GRP levels were significantly higher in the OA group (419.61 ± 70.14 ng/mL) compared to the non-OA group (382.18 ± 62.34 ng/mL; *p* = 0.037). MGP levels were significantly higher in the OA group (67.76 ± 11.36 ng/mL) compared to the non-OA group (53.49 ± 18.28 ng/mL; *p* = 0.001). Calcium levels (Ca^++^) were also significantly higher in the OA group (12.89 ± 3.43 mg/dL) compared to the non-OA group (9.51 ± 2.15 mg/dL; *p* = 0.0001). There was a significantly positive correlation between MGP levels and age (*p* = 0.011, R = +0.335). Linear regression analysis was performed to determine the effect of age on MGP levels (*p* = 0.011, R-Square = 0.112). The dependent variable in this analysis was MGP (ng/mL), and age was the predictor. **Conclusions:** In conclusion, both GRP and MGP are potentially usable biomarkers in osteoarthritis. However, GRP seems to be more valuable because it is not associated with age. In the future, both proteins could provide important contributions to the diagnosis and treatment of osteoarthritis.

## 1. Introduction

Osteoarthritis (OA) is a prevalent degenerative joint disease, especially among elderly individuals, and is a leading cause of physical disability worldwide. It significantly contributes to increased healthcare costs and impaired quality of life, particularly in aging populations [1,2]. Despite its prevalence, the pathophysiology of OA remains largely elusive. While mechanical wear and tear on the joints have long been considered the primary cause, growing evidence suggests that OA is a multifactorial disease influenced by genetic, metabolic, and inflammatory factors [3,4]. Genetic predisposition, obesity, joint malalignment, and aging are well-established risk factors for OA [5,6,7,8]. These factors may contribute to the degeneration of articular cartilage, subchondral bone remodeling, and synovial inflammation, all of which are hallmarks of OA. Additionally, recent studies have implicated pathologic calcification, crystal formation, and abnormal mineralization in the joint cartilage as key processes that may drive the progression of OA [9,10]. However, the role of cartilage mineralization in patients with OA is not completely understood.

Matrix Gla protein (MGP) and Gla-rich protein (GRP) are two vitamin-K-dependent proteins that have garnered attention for their roles in inhibiting calcification in various tissues, including cartilage. These proteins undergo post-translational modification, where specific glutamic acid (Glu) residues are converted into calcium-binding γ-carboxyglutamic acid (Gla) residues, a process dependent on vitamin K [11,12]. MGP, which is primarily synthesized by chondrocytes and vascular smooth-muscle cells, is a well-established inhibitor of vascular calcification and has been implicated in maintaining cartilage integrity [13]. GRP, on the other hand, is a more recently discovered Gla protein, and its functions are still being elucidated. Emerging evidence suggests that GRP, similar to MGP, may play a crucial role in preventing pathological calcification in the extracellular matrix [12]. MGP and GRP are indeed extracellular matrix proteins that can be secreted by cells. MGP is primarily synthesized by chondrocytes and vascular smooth-muscle cells, while GRP is also secreted by chondrocytes as well as other cell types within the joint environment. Both proteins play significant roles in the regulation of calcification processes. Upon secretion, these proteins can accumulate in the extracellular matrix and can be released into the synovial fluid, especially in pathological conditions, where tissue turnover and degradation are enhanced [14]. Although MGP has been studied extensively in the context of vascular calcification and cartilage health, research on GRP is still in its infancy. A few studies have explored the role of GRP in human tissues, with some suggesting that GRP may modulate the expression of proteins involved in calcification, such as α-smooth-muscle actin (α-SMA) and osteopontin (OPN) [15,16,17,18]. A study explored GRP’s role in inhibiting calcification within human aortic tissue. It demonstrated that GRP could increase α-smooth-muscle actin (αSMA) and downregulate osteopontin (OPN), highlighting its potential as a therapeutic agent in preventing vascular calcification [19]. Research indicated that GRP levels are associated with vascular calcification in chronic kidney disease (CKD) patients. Lower GRP levels correlated with the progression of CKD and increased vascular calcification, suggesting its utility as an early marker for vascular damage in these patients [16].

Considering the high calcium-binding affinity of Gla residues in both MGP and GRP, these proteins are thought to be key players in the regulation of connective tissue mineralization. This is particularly relevant in OA, where pathological mineralization of cartilage may contribute to disease progression. Despite the growing interest in the role of Gla proteins in calcification, the involvement of MGP and GRP in OA remains underexplored. [11,15]. Since Gla proteins have a very high affinity to Ca^++^ and pathologic mineralization is thought to be an important factor in OA, our hypothesis was that “the levels of GRP and MGP in the synovial fluid are expected to be significantly elevated in osteoarthritis”.

In this study, we aimed to compare the levels of MGP and GRP levels in the synovial fluid of osteoarthritic and non-osteoarthritic knee joints.

## 2. Materials and Methods

### 2.1. Study Design and Study Population

The study was conducted in accordance with the Declaration of Helsinki and approved by the Institutional Review Board of Ankara Training and Research Hospital (Approval: 2019/11) for studies involving humans. Informed consent was obtained from all participants in this double-blind, randomized clinical trial, conducted between January 2020 and November 2022 at Ankara Diskapi Yildirim Beyazit Education and Research Hospital. Knee synovial fluids were collected from patients treated in the Orthopedics and Traumatology Department, and the ELISA analysis was performed in the Medical Biology Department.

Two groups were established: the OA group and the non-OA group, as a control group. The non-OA group consisted of 31 patients (15 females, 16 males; mean age: 47.70 ± 14.49) who had no cartilage issues and underwent 25 partial meniscectomies and 6 anterior cruciate ligament reconstructions arthroscopically. The OA group included 26 patients (19 females, 7 males; mean age: 62.03 ± 11.53), all of whom underwent total knee arthroplasty because of grade 4 primary degenerative osteoarthritis. Patients with rheumatoid arthritis, traumatic septic arthritis osteoarthritis, or other inflammatory arthritis were excluded from the study.

At least 1 mL of knee synovial fluid was collected without lavage from both groups. In the OA group, synovial fluid was obtained from the joint space using a 5cc sterile syringe after arthrotomy. In the non-OA group, synovial fluid was collected by attaching a 10cc sterile syringe to the arthroscopic sheath and applying negative pressure after its insertion into the knee joint.

### 2.2. Measurements of GRP and MGP

Levels of GRP and MGP were quantified using commercial ELISA kits (Uscn Inc., Wuhan, China), with detectable ranges for GRP and MGP of 3.12–200 ng/mL and 80–0.125 ng/mL, respectively. Before performing the assay, the total protein content in each synovial fluid sample was quantified using the BCA Protein Assay Kit (Thermo Fisher Scientific, Waltham, MA, USA) to ensure consistent protein loading across all samples. This quantification step allowed for normalization of GRP and MGP levels relative to total protein content, enhancing the accuracy and reliability of the results.

The synovial fluid samples were diluted 1:15 with phosphate-buffered saline (PBS) to bring the concentrations of GRP and MGP within the detectable range of the assay. Following the ELISA procedure, all results were multiplied by a factor of 15 to account for this dilution. Positive and negative controls were included in each assay to validate the ELISA’s performance. The positive control consisted of a known concentration of recombinant GRP and MGP, while the negative control used PBS instead of synovial fluid to ensure no nonspecific binding or false positives occurred.

For patients with GRP and MGP levels exceeding the maximum detection limit of the assay, values were capped at 3000 ng/mL for GRP and 1200 ng/mL for MGP, as per the ELISA kit’s protocol. The lowest measurable concentrations for human GRP and MGP were less than 1.23 ng/mL and 50 pg/mL, respectively.

### 2.3. Statistical Analysis

Statistical analysis was conducted using SPSS software, version 27.0 (SPSS Inc., Chicago, IL, USA). Results were presented as mean values ± standard deviations and proportions. To assess the normality of the distribution of continuous variables, the Kolmogorov–Smirnov test was applied. This test is particularly useful for determining whether data significantly deviate from a normal distribution, which is a key assumption for many parametric tests [20]. In cases where the data followed a normal distribution, parametric tests were used; otherwise, appropriate non-parametric methods were considered. Categorical variables were analyzed using the chi-square test, which evaluates the association between categorical variables by comparing the observed frequencies in each category to the expected frequencies [21]. For comparisons of mean differences between two independent groups, the Student’s *t*-test was employed, which is appropriate for normally distributed data and helps determine whether there are statistically significant differences between the means of two groups [22]. The Pearson correlation coefficient was used to evaluate the strength and direction of the linear relationship between continuous variables. This coefficient provides a measure of the correlation between two variables, with values ranging from −1 to +1, where values closer to ±1 indicate a stronger linear relationship [23]. To assess the effect of age on MGP protein levels, linear regression analysis was conducted. Linear regression allows for the modeling of the relationship between a dependent variable and one or more independent variables, providing insights into the degree to which the independent variable(s) predict changes in the dependent variable [5]. In this analysis, age was treated as the independent variable, and MGP levels as the dependent variable. The significance of the regression model and the individual predictors was evaluated using *p*-values and confidence intervals [24]. A *p*-value of <0.05 was considered statistically significant for all analyses, indicating that the results were unlikely to have occurred by chance. All statistical tests were conducted following established guidelines and best practices in statistical analysis to ensure the reliability and validity of the findings.

## 3. Results

Demographic data and levels of specific biomarkers (GRP, MGP, and Ca^++^) in the synovial fluid of patients with osteoarthritis, compared with those without osteoarthritis, are shown in Table 1. There was no statistically significant difference in gender distribution between the two (*p* = 0.058). The mean age in the OA group (62.03 ± 11.53 years) was significantly higher than the non-OA group (47.70 ± 14.49 years; *p* = 0.0001). Regarding the biomarkers, GRP levels were 419.61 ± 70.14 ng/mL in the OA group and 382.18 ± 62.34 ng/mL in the non-OA group, with a *p*-value of 0.037, indicating a significant difference. MGP levels were significantly higher in the OA group (67.76 ± 11.36 ng/mL) compared to the non-OA group (53.49 ± 18.28 ng/mL), with a *p*-value of 0.001. Calcium levels (Ca^++^) were also significantly higher in the OA group (12.89 ± 3.43 mg/dL) compared to the non-OA group (9.51 ± 2.15 mg/dL), with a *p*-value of 0.0001. There was no correlation between GRP and MGP levels in all subjects (*p* = 0.875). There was a significantly positive correlation between MGP levels and age (*p* = 0.011, R = +0.335; Table 1 and Figure 1).

The linear regression analysis for determining the effect of age on MGP levels is shown in Table 2. Linear regression analysis was performed to determine the effect of age on MGP levels (*p* = 0.011, R-Square = 0.112). The dependent variable in this analysis was MGP (ng/mL), and age was the predictor. On the other hand, there was no correlation between GRP levels and age in all patients (*p* = 0.486; Table 2 and Table 3, and Figure 2).

## 4. Discussion

The results of this study showed that GRP and MGP levels significantly increased in osteoarthritic knees. We also found that MGP was correlated with aging, whereas GRP was not. In other words, the specificity of GRP was found to be higher than MGP.

The mechanism of osteoarthritis at the molecular level is still not completely understood. In some studies, matrix calcification and chondrocyte hypertrophy have been reported to be associated with the pathophysiology of OA [10,25,26,27,28]. Although some studies have reported that pathologic calcification is the primary effect of aging rather than OA [29], in contrast, there are also reports that concluded no correlation between the degree of mineralization and age [30,31].

Fuerst et al. evaluated the cartilage of 100 OA patients who had undergone total knee arthroplasty. They reported that cartilage mineralization was seen in all specimens. There was a relationship between mineralization and the expression of type X collagen, which is a marker for chondrocyte hypertrophy [30]. Also, a correlation was found between the degree of mineralization and knee scores, whereas no association was found between age and OA [30].

There are human [32,33] and animal [34,35] studies reporting associations between chondrocyte hypertrophy and OA. The standard marker of chondrocyte hypertrophy is type X collagen [36,37,38,39]. This collagen is frequently released from chondrocytes in OA, while it is not released from healthy cartilage [36,37]. The levels of collagenase-3 or matrix metalloprotease 13 (MMP13) are also increased in chondrocyte hypertrophy [40,41]. Recently, osteopontin, osteocalcin, Indian Hedgehog, Runx2, VEGF, HtrA1, and transglutaminase-2 (TG-2) were reported to be associated with chondrocyte hypertrophy [42,43].

GRP and MGP are vitamin-K-dependent proteins that contain Gla residues. Gla residues are well-known mineralization inhibitors and have an excessive affinity against Ca^++^ minerals. Unlike the other members of vitamin-K-dependent proteins, GRP can be expressed from all chondrocytes and other bone cells, such as osteocytes and osteoblasts [9,11,12,16,19].

MGP has been known for a long time, while GRP was recently discovered and, thus, very few data are available. GRP includes the highest number of Gla residues compared with the other members of vitamin-K-dependent proteins. For this reason, it is named Gla-rich protein [12]. In human [15] and animal studies [14,44], it was reported that GRP is most frequently expressed from chondrocyte cells. To the best of our knowledge, there are no previous reports showing the relationship between GRP and OA. However, increased expressions of GRP are reported in other pathologies, such as human dermatomyositis with calcinosis, pseudoxanthoma elasticum, calcified scar tissue, and chronic kidney disease related with arterial calcification [12]. Some in vitro studies have shown that during differentiation of a chondrocyte subclone, GRP expression largely parallels the expression of type II collagen, and decreases with maturation toward hypertrophic cells. It was also demonstrated that GRP expression was reversely correlated with the expression of type X collagen [36,37,38,39]. GRP has already been used as a marker in chondrocyte differentiation and chondrogenesis [45,46]. In light of the information presented above on the accumulation of GRP in sites of pathological calcifications, its relationship with fibrillar collagens suggests that GRP may play a key role in the pathogenesis of osteoarthritis. The results of our study showed that GRP levels in the synovial fluid significantly increased in osteoarthritis, and no correlation was found between age and GRP. This makes GRP valuable in terms of specificity.

MGP, a vitamin-K-dependent protein, is crucial for inhibiting soft tissue mineralization [11,14]. Luo et al. found that mice lacking MGP survive to term but die within two months due to arterial calcification, leading to blood-vessel rupture. They also observed inappropriate calcification in various cartilages, including the growth plate, resulting in short stature, osteopenia, and fractures [47]. Vitamin K is essential for the post-translational modification of Glu residues into Ca^++^-binding γ-carboxyglutamic acid residues. Wallin et al. compared the levels of the mature, fully γ-carboxylated form of MGP (cMGP) and non-γ-carboxylated MGP (ucMGP) in osteoarthritic (OA) and healthy cartilage. Their study revealed that osteoarthritic tissue produced significantly lower levels of cMGP compared to normal cartilage, suggesting that vitamin K deficiency might contribute to pathological mineralization in OA [48]. Although the exact mechanism of MGP is not fully understood, current research indicates that it may play a significant role in the pathogenesis of OA.

As far as we could find, there are limited studies in the literature regarding the relationship between OA and MGP [49,50,51,52]. To the best of our knowledge, the MGP level in the synovial fluid of osteoarthritic knees has not been studied previously. Our study demonstrated that MGP levels significantly increased in the synovial fluid of patients with OA. Age was found to be correlated with MGP levels. This weakened the specificity of MGP. However, we think that the results of our study support the idea that “MGP may play an important role in the molecular mechanism of osteoarthritis”.

### Limitations of the Study

The primary limitation of this study is that all patients with osteoarthritis (OA) included in the research had grade 4 OA. This focus on the most severe stage of OA may limit the generalizability of our findings to patients with less advanced stages of the disease. As a result, it remains unclear how the levels of Gla-rich protein (GRP) and matrix Gla protein (MGP) in synovial fluid might correlate with the severity of OA in earlier stages. To address this limitation, future research should aim to include a broader spectrum of OA patients, encompassing grades 1, 2, and 3. Such studies would provide a more comprehensive understanding of the relationship between GRP and MGP levels and the progression of OA across different stages of the disease. Additionally, longitudinal studies following patients over time could help clarify whether changes in these protein levels can serve as early biomarkers for disease progression. Further research could also explore the potential therapeutic implications of modulating GRP and MGP levels in the synovial fluid, potentially opening new avenues for the treatment of OA. Investigating the molecular mechanisms by which these proteins influence cartilage health and joint function could offer insights into novel strategies for preventing or slowing the progression of OA. By expanding the study to include patients with varying degrees of OA severity and pursuing these additional research directions, we hope to overcome the current limitations and contribute to a deeper understanding of the role of vitamin-K-dependent proteins in the pathophysiology of OA.

## 5. Conclusions

GRP and MGP are both vitamin-K-dependent proteins that play an important role in the inhibition of mineralization. As mentioned above, pathologic mineralization is one of the main findings of osteoarthritis. In our opinion, the expression of these proteins increases in response to the high level of Ca^++^ ions in the synovial fluid. Also, we think that after the increased expression of GRP and MGP proteins, they bind Ca^++^ ions in order to prevent pathologic mineralization. Further advanced studies are needed to highlight the mechanism.

In conclusion, both GRP and MGP are potentially usable biomarkers in OA. However, GRP seems to be more valuable because it is not associated with age. In the future, they could provide important contributions to the diagnosis and treatment of OA.

## Figures and Tables

**Figure 1 jcm-13-05159-f001:**
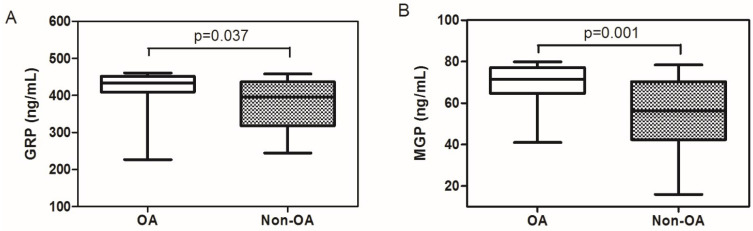
The comparison of GRP (**A**) and MGP (**B**) levels between OA and non-OA groups. OA: osteoarthritis; non-OA: non-osteoarthritis.

**Figure 2 jcm-13-05159-f002:**
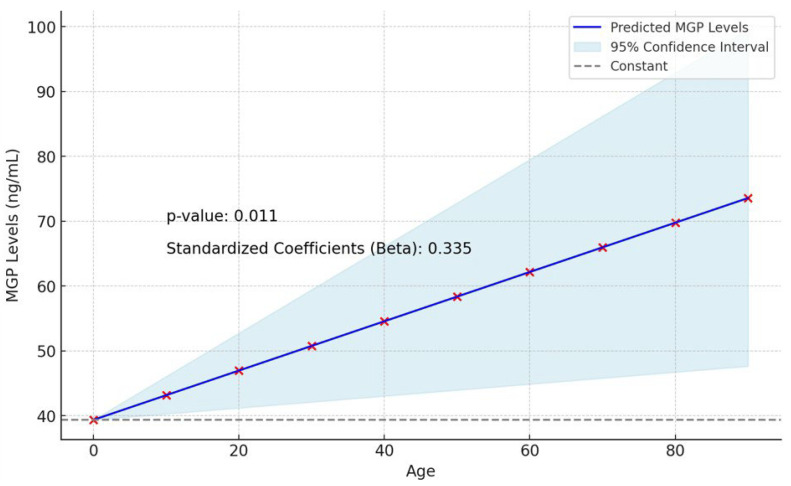
Linear regression analysis: effects of age on MGP levels.

**Table 1 jcm-13-05159-t001:** Demographic data of the patients and levels of GRP, MGP, and Ca^++^ in the synovial fluids.

	OA (*n* = 26) Mean ± SD	Non-OA (*n* = 31) Mean ± SD	*p*-Value
Gender			
Male (*n*)	7 (26.9%)	16 (51.6%)	0.058 *
Female (*n*)	19 (73.1%)	15 (48.4%)	
Age (years)	62.03 ± 11.53	47.70 ± 14.49	0.0001 **
GRP (ng/mL)	419.61 ± 70.14	382.18 ± 62.34	0.037 **
MGP (ng/mL)	67.76 ± 11.36	53.49 ± 18.28	0.001 **
Ca^++^ (mg/dL)	12.89 ± 3.43	9.51 ± 2.15	0.0001 **

Chi-square tests *, independent-samples test **, and SD = standard deviation.

**Table 2 jcm-13-05159-t002:** Linear regression analysis for determining the effect of age on MGP levels.

	Unstandardized Coefficients		Standardized Coefficients	t	*p*	95.0% Confidence Interval for B	
	B	Std. Error	Beta			Lower Bound	Upper Bound
Constant	39.368	8.107		4.856	0.000	23.121	55.615
Age	0.380	0.144	0.335	2.639	0.011	0.092	0.669

Dependent variable: MGP (ng/mL), predictor: age, and Durbin–Watson = 1.690.

**Table 3 jcm-13-05159-t003:** Linear regression analysis for determining the effect of age on GRP levels.

	Unstandardized Coefficients		Standardized Coefficients	t	*p*	95.0% Confidence Interval for B	
	B	Std. Error	Beta			Lower Bound	Upper Bound
Constant	30.000	7.500		4.000	0.000	15.140	45.520
Age	0.100	0.140	0.080	0.700	0.486	−0.0180	0.380

Dependent variable: GRP (ng/mL), predictor: age, and Durbin–Watson = 1.750.

## Data Availability

The original contributions presented in the study are included in the article, further inquiries can be directed to the corresponding authors.

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
