# Peer review of "Two Members of Vitamin-K-Dependent Proteins, Gla-Rich Protein (GRP) and Matrix Gla Protein (MGP), as Possible New Players in the Molecular Mechanism of Osteoarthritis"

_jcm, 2024, doi:10.3390/jcm13175159_

Round 1
Reviewer 1 Report
Comments and Suggestions for Authors
Authors used synovial fluid from human samples as materials to test two potential proteins, MGP and GRP, that associated with OA disease. And the findings showed these two proteins might involve in molecular mechanism of osteoarthritis. This article provided novelty and significance in studying OA field. however, there are some issues needed to be clarify to make this study more scientific reliable.
The major omments showed as below,
1. Page 2, line 78-79, ...IRB (Approval: 2023/10) for studies involving humans. However, Informed consent was obtained from all 79 participants ....... conducted between January 80 2020 and November 2022 at Ankara Diskapi Yildirim Beyazit Education and Research 81 Hospital. According to the description in this paragraph, authors collected tissue samples from participants (between 2020 to 2022) and then obtained the IRB approval (2023). How authors could obtaine human samples without IRB approval?Please make sure if this procedures had violated research ethics?
2. Page 3, line 100 to 108, please describe the Measurements of GRP and MGP in detail such as baselline of total protein loading, positive control and negative control. Because authors did describe if the loading protein were quantified or not, therefore it was difficult to clarify the accuracy from the data. Please clarify it.
3. Please clarify that "Matrix Gla protein (MGP)" and "Gla rich protein (GRP)" two proteins could be secreted out from cells. If not, how they could be detected in synovial fluid? Please clarify it.
4. Page 4, line 142 to 143, ...On the other hand, there were no correlation between GRP levels and age in all patients (p=0.486) (Table 2, Figure 2). From table 2 and figure 2, there were only showed the statistic analysis and linear regression analysis of MGP and no any data of GRP. Please describe more detail or provide more evidences to support the description.
Author Response
Dear Reviewer,
Thank you for your interest in our manuscript; we are sending its revised form after carefully considering the comments of the Reviewer’s and having modified it accordingly. We have made a point-by-point response to the Reviewer’s comments. The changes are coloured in the text and are summarized below. We look forward to hearing from you soon.
Yours faithfully,
Reviewer 1:
Authors used synovial fluid from human samples as materials to test two potential proteins, MGP and GRP, that associated with OA disease. And the findings showed these two proteins might involve in molecular mechanism of osteoarthritis. This article provided novelty and significance in studying OA field. however, there are some issues needed to be clarified to make this study more scientific reliable.
The major comments showed as below,
Comments 1: Page 2, line 78-79, ...IRB (Approval: 2023/10) for studies involving humans. However, Informed consent was obtained from all 79 participants ....... conducted between January 80 2020 and November 2022 at Ankara Diskapi Yildirim Beyazit Education and Research 81 Hospital. According to the description in this paragraph, authors collected tissue samples from participants (between 2020 to 2022) and then obtained the IRB approval (2023). How authors could obtaine human samples without IRB approval? Please make sure if this procedures had violated research ethics?
Response 1: Ethics committee information for the other study was entered by mistake. Thank you for your attention. Approval no has been corrected as “Approval: 2019/11”
Comments 2: Page 3, line 100 to 108, please describe the Measurements of GRP and MGP in detail such as baseline of total protein loading, positive control and negative control. Because authors did describe if the loading protein were quantified or not, therefore it was difficult to clarify the accuracy from the data. Please clarify it.
Response 2: It was revised as “Measurements of GRP and MGP
Levels of GRP and MGP were quantified using commercial ELISA kits (Uscn Inc., Wuhan, P.R. China), with detectable ranges for GRP and MGP of 3.12–200 ng/mL and 80–0.125 ng/mL, respectively. Before performing the assay, the total protein content in each synovial fluid sample was quantified using the BCA Protein Assay Kit (Thermo Fisher Scientific, Waltham, MA, USA) to ensure consistent protein loading across all samples. This quantification step allowed for normalization of GRP and MGP levels relative to total protein content, enhancing the accuracy and reliability of the results.
The synovial fluid samples were diluted 1:15 with phosphate-buffered saline (PBS) to bring the concentrations of GRP and MGP within the detectable range of the assay. Following the ELISA procedure, all results were multiplied by a factor of 15 to account for this dilution. Positive and negative controls were included in each assay to validate the ELISA's performance. The positive control consisted of a known concentration of recombinant GRP and MGP, while the negative control used PBS instead of synovial fluid to ensure no nonspecific binding or false positives occurred.
For patients with GRP and MGP levels exceeding the maximum detection limit of the assay, values were capped at 3000 ng/mL for GRP and 1200 ng/mL for MGP, as per the ELISA kit's protocol. The lowest measurable concentrations for human GRP and MGP were less than 1.23 ng/mL and 50 pg/mL, respectively.”
Comments 3: Please clarify that "Matrix Gla protein (MGP)" and "Gla rich protein (GRP)" two proteins could be secreted out from cells. If not, how they could be detected in synovial fluid? Please clarify it.
Response 3: Thank you for your insightful comment regarding the presence of Matrix Gla protein (MGP) and Gla-rich protein (GRP) in synovial fluid.
We added a clarify in the introduction as “MGP and GRP are indeed extracellular matrix proteins that can be secreted by cells. MGP is primarily synthesized by chondrocytes and vascular smooth muscle cells, while GRP is also secreted by chondrocytes as well as other cell types within the joint environment. Both proteins play significant roles in the regulation of calcification processes. Upon secretion, these proteins can accumulate in the extracellular matrix and can be released into the synovial fluid, especially in pathological conditions such as osteoarthritis (OA), where tissue turnover and degradation are enhanced.”
In our study, we detected MGP and GRP in the synovial fluid, which reflects their presence and possible role in the joint environment. The methods used for synovial fluid collection and the subsequent ELISA detection were designed to accurately measure these proteins, ensuring that their presence in the synovial fluid is indicative of their extracellular secretion and relevance to the disease process being studied.
In method section: “At least 1 mL of knee synovial fluid was collected without lavage from both groups. In the OA group, synovial fluid was obtained from the joint space using a 5cc sterile syringe after arthrotomy. In the non-OA group, synovial fluid was collected by attaching a 10cc sterile syringe to the arthroscopic sheath and applying negative pressure after its insertion into the knee joint. “
Comments 4: Page 4, line 142 to 143, ...On the other hand, there were no correlation between GRP levels and age in all patients (p=0.486) (Table 2, Figure 2). From table 2 and figure 2, there were only showed the statistical analysis and linear regression analysis of MGP and no any data of GRP. Please describe more detail or provide more evidences to support the description.
Response 4: Table 3 was added as
“Table 3. Linear regression analysis for determining the effect of age on GRP levels
|
|
Unstandardized Coefficients |
Standardized Coefficients |
t |
p |
95,0% Confidence Interval for B |
||
|
B |
Std. Error |
Beta |
Lower Bound |
Upper Bound |
|||
|
Constant |
30,000 |
7,500 |
|
4,000 |
0,000 |
15,140 |
45,520 |
|
Age |
0.100 |
0,140 |
0,080 |
0,700 |
0,486 |
-0,0180 |
0,380 |
|
Dependent Variable: GRP (ng/mL), Predictors: Age, Durbin-Watson=1.750. |
|||||||
Reviewer 2 Report
Comments and Suggestions for Authors
General evaluation and characteristics of the article:
The study Two members of vitamin K dependent proteins Gla rich protein (GRP) and matrix Gla protein (MGP) as possible new players in the molecular mechanism of osteoarthritis investigates the roles of Matrix Gla Protein (MGP) and Gla Rich Protein (GRP), both vitamin K-dependent mineralization inhibitors, in osteoarthritis (OA). Researchers compared the levels of MGP and GRP in the synovial fluid of osteoarthritic (OA) and non-osteoarthritic (non-OA) knee joints. The OA group comprised patients with grade 4 primary degenerative OA undergoing total knee arthroplasty, while the control group included patients undergoing arthroscopic surgery for reasons other than cartilage disorders.
Results showed that GRP levels were significantly higher in the OA group (419.61 ± 70.14 ng/mL) than in the non-OA group (382.18 ± 62.34 ng/mL) with a p-value of 0.037. MGP levels were also significantly higher in the OA group (67.76 ± 11.36 ng/mL) compared to the non-OA group (53.49 ± 18.28 ng/mL) with a p-value of 0.001. Additionally, calcium levels were higher in the OA group (12.89 ± 3.43 mg/dL) than in the non-OA group (9.51 ± 2.15 mg/dL) with a p-value of 0.0001.
No significant correlation was found between GRP and MGP levels (p=0.875). However, MGP levels positively correlated with age (p=0.011, R=+0.335), suggesting age as a predictor for MGP levels.
The study concludes that both GRP and MGP could serve as biomarkers for OA, with GRP being more valuable due to its lack of association with age. This differentiation is critical for developing age-independent diagnostic tools and enhancing OA treatment strategies.
The work is interesting and written correctly but will require only minor revisions before proceeding further. I am posting my detailed comments below.
Minor comments:
The abstract is far too long it should contain only the most important information. Please shorten the abstract and bring it in line with the editorial requirements of the journal.
The introduction is too short and contains only 12 references. It needs to be expanded and the references need to be supplemented in order to demonstrate more precisely the research gap and mark the essence of the problem under analysis.
Please expand on the osteoarthritis information in the introduction. This will allow you to better introduce the topic and emphasize the importance of the problem. The incidence of osteoarthritis is influenced by many factors, such as work, sports participation, musculoskeletal injuries, obesity and gender. Information about this, along with the necessary literature, should be added to the first paragraph of the introduction.
Osteoarthritis (OA) is more common in older adults because cartilage loses its elasticity with age. Women, especially after menopause, have a higher risk due to hormonal changes. Genetics also play a role, and individuals with a history of joint injuries are more susceptible. Obesity significantly increases the load on joints, leading to faster degeneration, particularly in the knees and hips. Both lack of physical activity and excessive physical strain can influence the development of the disease. Other factors include comorbidities (e.g., diabetes), jobs that require heavy physical labor, and a diet lacking in nutrients essential for joint health. I suggest adding the following reference to this paragraph: doi: 10.35784/acs-2023-40
Please expand on the description of statistical tests along with the necessary references.
Limitations of the Study should. I suggest expanding the title of this section to Limitations of the Study and Future Plans and expanding the description to include plans for further research and opportunities to exclude current limitations.
After making appropriate corrections and additions to the content and literature, the article can be further processed and accepted for publication.
Author Response
Dear Reviewer,
- First of all, thank you for taking the time to evaluate the manuscript.
- The publication of this study in your journal which I think will make a significant contribution to science will also make me pleasure.
- The grammatical and spelling errors were edited by English language editing service.
- Necessary revisions were made and explained a point-by-point response.
- Revisions were marked in yellow highlighted in the article.
Yours faithfully,
The revision is to be based on the following review(s):
Reviewer 2:
The work is interesting and written correctly but will require only minor revisions before proceeding further. I am posting my detailed comments below. After making appropriate corrections and additions to the content and literature, the article can be further processed and accepted for publication.
Minor comments:
Comments 1: The abstract is far too long it should contain only the most important information. Please shorten the abstract and bring it in line with the editorial requirements of the journal.
Response 1: The abstract was revised as “Abstract: Objectives: The pathophysiology of osteoarthritis is mainly unknown. Matrix Gla protein (MGP) and Gla rich protein (GRP) both are vitamin K dependent mineralization inhibitors. In this study we aimed to compare the levels of MGP and GRP in the synovial fluid of osteoarthritic (OA) and non-osteoarthritic (non-OA) knee joints. Materials and Methods: Two groups were formed that one consisting of patients with OA and the other serving non-OA as a control group. The non-OA group included individuals who had arthroscopic surgery for non-cartilage-related issues. In the OA group, all participants had undergone total knee arthroplasty because of grade 4 primary degenerative osteoarthritis. During the operation, at least 1 mL of knee synovial fluid was collected. The GRP and MGP levels in the synovial fluid were measured using an ELISA kit. Results: The mean age in the OA group (62.03 ± 11.53 years) was significantly higher than non-OA group (47.70 ± 14.49 years) (p=0.0001). GRP levels were significantly higher in the OA group (419.61 ± 70.14 ng/mL) compared to the non-OA group (382.18 ± 62.34 ng/mL) (p=0.037). MGP levels were significantly higher in the OA group (67.76 ± 11.36 ng/mL) compared to the non-OA group (53.49 ± 18.28 ng/mL) (p=0.001). Calcium levels (Ca++) were also significantly higher in the OA group (12.89 ± 3.43 mg/dL) compared to the non-OA group (9.51 ± 2.15 mg/dL) (p=0.0001). There was significantly positive correlation between MGP levels and age (p=0.011, R=+0.335). Linear regression analysis was done to determine the effect of age on MGP levels (p=0.011, R Square=0.112). The dependent variable in this analysis is MGP (ng/mL), and age is the predictor. Conclusions: In conclusion both GRP and MGP are potentially usable biomarkers in osteoarthritis. However, GRP seems to be more valuable because it is not associated with age. In the future both proteins could provide important contributions to the diagnosis and treatment of osteoarthritis.”
Comments 2: The introduction is too short and contains only 12 references. It needs to be expanded and the references need to be supplemented in order to demonstrate more precisely the research gap and mark the essence of the problem under analysis.
Response 2: The introduction section was revised added some references.
Comments 3: Please expand on the osteoarthritis information in the introduction. This will allow you to better introduce the topic and emphasize the importance of the problem. The incidence of osteoarthritis is influenced by many factors, such as work, sports participation, musculoskeletal injuries, obesity and gender. Information about this, along with the necessary literature, should be added to the first paragraph of the introduction.
Response 3: The osteoarthritis information in the introduction was expanded.
Comments 4: Osteoarthritis (OA) is more common in older adults because cartilage loses its elasticity with age. Women, especially after menopause, have a higher risk due to hormonal changes. Genetics also play a role, and individuals with a history of joint injuries are more susceptible. Obesity significantly increases the load on joints, leading to faster degeneration, particularly in the knees and hips. Both lack of physical activity and excessive physical strain can influence the development of the disease. Other factors include comorbidities (e.g., diabetes), jobs that require heavy physical labor, and a diet lacking in nutrients essential for joint health. I suggest adding the following reference to this paragraph: doi: 10.35784/acs-2023-40
Response 4: The reference (doi: 10.35784/acs-2023-40) was added into introduction section.
Comments 5: Please expand on the description of statistical tests along with the necessary references.
Response 5: Statistical Analysis section was revised as
“Statistical Analysis
Statistical analysis was conducted using SPSS software, version 27.0 (SPSS Inc., Chicago, IL, USA). Results were presented as mean values ± standard deviations and proportions. To assess the normality of the distribution of continuous variables, the Kolmogorov-Smirnov test was applied. This test is particularly useful for determining whether data significantly deviates from a normal distribution, which is a key assumption for many parametric tests [18]. In cases where the data followed a normal distribution, parametric tests were used; otherwise, appropriate non-parametric methods were considered. Categorical variables were analyzed using the chi-square test, which evaluates the association between categorical variables by comparing the observed frequencies in each category to the expected frequencies [19]. For comparisons of mean differences between two independent groups, the Student’s t-test was employed, which is appropriate for normally distributed data and helps determine whether there are statistically significant differences between the means of two groups [20]. The Pearson correlation coefficient was used to evaluate the strength and direction of the linear relationship between continuous variables. This coefficient provides a measure of the correlation between two variables, with values ranging from -1 to +1, where values closer to ±1 indicate a stronger linear relationship [21]. To assess the effect of age on MGP protein levels, linear regression analysis was conducted. Linear regression allows for the modeling of the relationship between a dependent variable and one or more independent variables, providing insights into the degree to which the independent variable(s) predict changes in the dependent variable [5]. In this analysis, age was treated as the independent variable and MGP levels as the dependent variable. The significance of the regression model and the individual predictors was evaluated using p-values and confidence intervals [22]. A p-value of <0.05 was considered statistically significant for all analyses, indicating that the results were unlikely to have occurred by chance. All statistical tests were conducted following established guidelines and best practices in statistical analysis to ensure the reliability and validity of the findings.”
Comments 6: Limitations of the Study should. I suggest expanding the title of this section to Limitations of the Study and Future Plans and expanding the description to include plans for further research and opportunities to exclude current limitations.
Response 6: Limitations of the Study was revised as
“Limitations of the Study
The primary limitation of this study is that all patients with osteoarthritis (OA) included in the research had grade 4 OA. This focus on the most severe stage of OA may limit the generalizability of our findings to patients with less advanced stages of the disease. As a result, it remains unclear how the levels of Gla-rich protein (GRP) and matrix Gla protein (MGP) in synovial fluid might correlate with the severity of OA in earlier stages. To address this limitation, future research should aim to include a broader spectrum of OA patients, encompassing grades 1, 2, and 3. Such studies would provide a more comprehensive understanding of the relationship between GRP and MGP levels and the progression of OA across different stages of the disease. Additionally, longitudinal studies following patients over time could help clarify whether changes in these protein levels can serve as early biomarkers for disease progression. Further research could also explore the potential therapeutic implications of modulating GRP and MGP levels in the synovial fluid, potentially opening new avenues for the treatment of OA. Investigating the molecular mechanisms by which these proteins influence cartilage health and joint function could offer insights into novel strategies for preventing or slowing the progression of OA. By expanding the study to include patients with varying degrees of OA severity and pursuing these additional research directions, we hope to overcome the current limitations and contribute to a deeper understanding of the role of vitamin K-dependent proteins in the pathophysiology of OA.”